# Multi-Task Deep Learning Model for Classification of Dental Implant Brand and Treatment Stage Using Dental Panoramic Radiograph Images

**DOI:** 10.3390/biom11060815

**Published:** 2021-05-30

**Authors:** Shintaro Sukegawa, Kazumasa Yoshii, Takeshi Hara, Tamamo Matsuyama, Katsusuke Yamashita, Keisuke Nakano, Kiyofumi Takabatake, Hotaka Kawai, Hitoshi Nagatsuka, Yoshihiko Furuki

**Affiliations:** 1Department of Oral and Maxillofacial Surgery, Kagawa Prefectural Central Hospital, 1-2-1, Asahi-machi, Takamatsu, Kagawa 760-8557, Japan; mtamamo@outlook.com (T.M.); furukiy@ma.pikara.ne.jp (Y.F.); 2Dentistry and Pharmaceutical Sciences, Department of Oral Pathology and Medicine, Okayama University Graduate School of Medicine, Okayama 700-8558, Japan; pir19btp@okayama-u.ac.jp (K.N.); gmd422094@s.okayama-u.ac.jp (K.T.); de18018@s.okayama-u.ac.jp (H.K.); jin@okayama-u.ac.jp (H.N.); 3Department of Intelligence Science and Engineering, Graduate School of Natural Science and Technology, Gifu University, 1-1 Yanagido, Gifu, Gifu 501-1193, Japan; yoshii@fjt.info.gifu-u.ac.jp (K.Y.); takeshi.hara@mac.com (T.H.); 4Center for Healthcare Information Technology, Tokai National Higher Education and Research System, 1-1 Yanagido, Gifu, Gifu 501-1193, Japan; 5Polytechnic Center Kagawa, 2-4-3, Hananomiya-cho, Takamatsu, Kagawa 761-8063, Japan; kazamakura.ka2suke@gmail.com

**Keywords:** multi-task learning, deep learning, artificial intelligence, dental implant, classification

## Abstract

It is necessary to accurately identify dental implant brands and the stage of treatment to ensure efficient care. Thus, the purpose of this study was to use multi-task deep learning to investigate a classifier that categorizes implant brands and treatment stages from dental panoramic radiographic images. For objective labeling, 9767 dental implant images of 12 implant brands and treatment stages were obtained from the digital panoramic radiographs of patients who underwent procedures at Kagawa Prefectural Central Hospital, Japan, between 2005 and 2020. Five deep convolutional neural network (CNN) models (ResNet18, 34, 50, 101 and 152) were evaluated. The accuracy, precision, recall, specificity, F1 score, and area under the curve score were calculated for each CNN. We also compared the multi-task and single-task accuracies of brand classification and implant treatment stage classification. Our analysis revealed that the larger the number of parameters and the deeper the network, the better the performance for both classifications. Multi-tasking significantly improved brand classification on all performance indicators, except recall, and significantly improved all metrics in treatment phase classification. Using CNNs conferred high validity in the classification of dental implant brands and treatment stages. Furthermore, multi-task learning facilitated analysis accuracy.

## 1. Introduction

Dental implants have been used for more than half a century and currently are a highly reliable treatment option for long-term replacement (10+ years) of missing teeth [1,2]. As a long-term prognosis-established treatment, incidences of mechanical complications, such as fractures on fixtures or abutment screws, and biological complications, such as peri-implantitis, will inevitably occur [3,4,5,6]. An accurate understanding of the efficacy of different implant brands and the identification of treatment status are important for continued implant maintenance management and for dealing with any complications that arise. Unfortunately, it has been reported that about 3% of implants must be removed without continuous prosthetic treatment or repair, solely due to the inability to identify the type of implant used [7]. Patients may not be able to visit the same dentist for several reasons, including poor general condition, treatment transfer, emigration, or closure of their dentist’s office. Consequently, the necessary information regarding the implants is unavailable. Therefore, it is very important to be able to independently and accurately identify the type of implant used in a patient.

Each brand of dental implant has a relatively distinctive morphology that can be used for proper identification. Brand identification, along with the accurate determination of treatment stage is crucial for efficient treatment and can be achieved through the use of medical images. Dental panoramic radiography is widely used in dental clinics and in dental or oral surgery. Because the entire tooth and jaw can be imaged at the same time, this method is useful for simultaneously capturing a large amount of information, such as the condition of the teeth and jawbone, prosthesis details, and the implant shape [8].

By the early 2000s, it was estimated that over 2000 different types of dental implants were available in the market [9]. The development and verification of a wide variety of fixture structures and implants is ongoing. Unfortunately, it is difficult for dentists to accurately identify certain implants if they have no experience with them.

Deep learning systems employ artificial intelligence machine learning techniques that allow computers to learn human-like tasks based on neural networks; therefore, they mimic neural circuitry in the human brain. The most frequently used computer technology in these types of research is deep learning using convolutional neural networks (CNNs). Deep learning using CNNs are very useful for classification and diagnosis when using medical images [10]. In recent years, research on deep learning models using panoramic radiographs has been reported, including tooth detection and numbering [11], osteoporosis prescreening [12], cystic lesion detection [13,14], atherosclerotic carotid plaque detection [15], and maxillary sinusitis diagnosis [16]. There are many different methods of machine learning; among them, multi-task learning is based on the theory that by learning interrelated concepts, classification methods can be developed through a wide range of generalizations and can ultimately improve performance compared to single-task learning [17].

In this study, we propose a novel approach for interpreting medical images while improving the generalization capabilities of multiple tasks. The purpose of this study was to build and evaluate a method that classifies implant brands and treatment stages from dental panoramic radiographic images using a multi-task deep learning approach.

## 2. Patients and Methods

### 2.1. Study Design

The purpose of this study was to classify implant brands and implant treatment stages from datasets segmented from dental panoramic radiographs using several Residual Neural Network (ResNet) CNNs Available online: https://github.com/qubvel/classification_models (accessed on 19th April 2021). Supervised learning was used as a method of deep learning. We also compared the multi-task and single-task accuracy of brand classification and implant treatment stage classification.

### 2.2. Ethics Statement

We retrospectively used radiographic data from January 2005 to December 2020. This study protocol was approved by the institutional review committee of Kagawa Prefectural Central Hospital (approval number 1019, approved on 8th March 2021). The institutional review committee waived the need for individual informed consent. Therefore, written/verbal informed consent was not obtained from any participant because this study featured a non-interventional retrospective design, and all data were analyzed anonymously.

### 2.3. Data Acquisition and Preprocessing

Dental panoramic radiographs of each patient were used to acquire images using AZ3000CMR and Hyper-G CMF (Asahiroentgen Ind. Co., Ltd., Kyoto, Japan). All data images were output in Tagged Image File Format (TIFF) format (2964 × 1464, 2694 × 1450, 2776 × 1450 or 2804 × 1450 pixels) from the Kagawa Prefectural Central Hospital Picture Archiving and Communication Systems (PACS) system (Hope Dr Able-GX, Fujitsu Co., Tokyo, Japan). The radiographic images were classified and labeled based on electronic medical records and the dental implant usage ledger of our department. From a collection of 7079 selected digital panoramic dental radiographs, a dataset of 9767 manually cropped image segments, each focused on a dental implant, was synthesized.

Each dental implant image included was manually cropped as needed for each dental panoramic radiograph taken. These dental implant data included implant fixtures, healing abutments, provisional settings, and final prostheses. For preparation before analysis, we used Photoshop Element (Adobe Systems, Inc., San Jose, CA, USA) to manipulate the images so that all dental implant fixtures would fit (see Figure 1 and Figure 2). The cropped image was saved in portable network graphics (PNG) format. Oral and maxillofacial surgeons who performed the cropping were completely unaware of the accurate implant brand of each patient.

The data were digitally preprocessed. Each image captured in PNG format was resized to 128 × 128 pixels. The preprocessing did not change the orientation of the image.

#### 2.3.1. Classification of Dental Implant Brand

The 12systems mainly used at Kagawa Prefectural Central Hospital were selected as the dental implants targeted in this study. The types of dental implant systems and corresponding number of images are shown in Table 1. Among them, images containing the following 12 types of dental implant systems were selected for this work:(1)**Full OSSEOTITE 4.0**: Full OSSEOTITE Tapered Certain (Zimmer Biomet, Palm Beach Gardens, FL, USA), diameter of 4 mm; lengths of 8.5, 10, 11 and 11.5 mm.(2)**Astra EV 4.2**: Astra Tech Implant System OsseoSpeed EV (Dentsply IH AB, Molndal, Sweden), diameter of 4.2 mm; lengths of 9 and 11 mm.(3)**Astra TX 4.0**: Astra Tech Implant System OsseoSpeed TX (Dentsply IH AB, Molndal, Sweden), diameter of 4 mm; lengths of 8, 9 and 11 mm.(4)**Astra TX 4.5**: Astra Tech Implant System OsseoSpeed TX (Dentsply IH AB, Molndal, Sweden), diameter of 4.5 mm; lengths of 9 and 11 mm.(5)**Astra MicroThread 4.0**: Astra Tech Implant System MicroThread, (Dentsply IH AB, Molndal, Sweden), diameter of 4 mm; lengths of 8, 9 and 11 mm.(6)**Astra MicroThread 4.5**: Astra Tech Implant System MicroThread, (Dentsply IH AB, Molndal, Sweden), diameter of 4.5 mm; lengths of 9 and 11 mm.(7)**Brånemark Mk III 4.0**: Brånemark System Mk III TiUnite (Nobelbiocare, Göteborg, Sweden), diameter of 4 mm; lengths of 8.5, 10 and 11.5 mm.(8)**FINESIA 4.2**: FINESIA BL HA TP (Kyocera Co., Kyoto, Japan), diameter of 4.2 mm; lengths of 8 and 10 mm.(9)**Replace Select Tapered 4.3**: Replace Select Tapered (Nobelbiocare, Göteborg, Sweden), diameter of 4.3 mm; lengths of 8, 10 and 11.5 mm.(10)**Nobel Replace CC 4.3**: Nobel Replace Conical Connection, (Nobelbiocare, Göteborg, Sweden), diameter of 4.3 mm; lengths of 8, 10 and 11.5 mm.(11)**Straumann Tissue 4.1**: Standard Plus Implant Tissue Level implants (Straumann Group, Basei, Switzerland), diameter of 4.1 mm; lengths of 8 and 10 mm.(12)**Straumann Bone Level 4.1**: Standard Plus Implant Bone Level implants (Straumann Group, Basei, Switzerland), diameter of 4.1 mm; lengths of 8 and 10 mm.

#### 2.3.2. Classification of Dental Implant Treatment Stages

The treatment stage is either the implant fixture after the primary surgery, the implant fixture after the implant placement by the secondary surgery, or the one-stage implant placement with the healing abutment attached, and the prosthetic set, including the final prosthesis and the provisional restoration setting. We included this task in this study because understanding the stage of treatment is clinically important. Different implant brands use different drivers for cover screws and healing abutments, and it is necessary to prepare the equipment according to the stage of treatment. Understanding the treatment stage and brand at the same time is essential for smooth treatment. These were classified into three categories. All classifications were based on clinical chart records.

### 2.4. CNN Model Architecture

In this study, the evaluation was performed using the standard CNN model ResNet [18]. ResNet was invented by He et al. [18]. It is generally accepted that the accuracy of image discrimination is improved by deepening the network layer; conversely, if the network layer is too deep, the accuracy will decrease. To deal with this, we introduced an already developed learning method called residual learning, which has the following advantage: its batch normalization solves the gradient disappearance and makes model deterioration less likely to occur [19]. Thus, ResNet is a network that can be deepened to very deep layers of over 100 layers. This representative of the ResNet architecture has layers 18, 34, 50, 101 and 152, which were selected as the CNN model in this study.

With efficient model construction, fine-tuning the weight of existing models as initial values for additional learning is possible; therefore, all CNNs were used to transfer learning with fine-tuning employed pre-trained weights using the ImageNet database [20]. The process of deep learning classification was implemented using Python language (version.3.7.10) and the Keras (version.2.4.3) Available online: https://github.com/keras-team/keras (accessed on 19th April 2021).

### 2.5. Model Training

The model training was generalized using k-fold cross-validation in the model training algorithm. The training algorithm used k = 4 for k-fold cross-validation to avoid overfitting and bias and to minimize generalization errors [21]. The data were divided into four and the test data consisted of 1950 images. Within each fold, the dataset was partitioned into independent training and validation sets, using an 80–20 percentage split. The selected validation set was a completely independent fold from the other training folds, and it was used to evaluate the training status during the training. After completing one model training step, we performed similar validations four times with different test data.

### 2.6. Deep Learning Procedure

All models were trained and evaluated on a 64-bit Ubuntu 16.04.5 LTS operating system with 8 GB memory and an NVIDIA GeForce GTX 1080 8 GB graphics processing unit. The optimizer, weight decay, and momentum were common to all the CNNs. In this study, the optimizer used stochastic gradient descent, and the weight decay and momentum were 0 and 0.9, respectively. Learning rates of 0.001 were used for ResNet for each. All the models analyzed a maximum of 50 epochs and minibatch size 32. We used the early stop method to terminate data training to prevent overfitting if the validation error did not update 10 times in a row.

### 2.7. Multi-Task

As a novel approach for the implant brand and treatment stage classifier, a deep neural network with two independent outputs was implemented and evaluated. The proposed multi-task CNN can analyze the implant brand and the treatment stage simultaneously. The model can reduce the number of trainable parameters that would otherwise be required when using two independent CNN models for implant brand and treatment stage classification. The proposed model has feature learning shared layers, including convolutional and polling layers that are shared with two separated branches, with independent, fully connected layers used for classification. The multi-task CNN consisted of ResNet18, 34, 50, 101 and 152. The convolution layer and pooling layer, excluding the FC layer from each ResNet architecture, were used for feature learning. For classification, two individual branches composed of dense layers were connected to single-output layers for the implant brand and the treatment stage, each one with softmax activations.

The scheme proposed for the implant assessment using the multi-task CNN models is shown in Figure 2. Table 2 shows the number of parameters for each multi-task and single-task model in ResNet.

In the multi-task model, learning about the classification of implant brands and learning about the classification of treatment stages are performed. For both learnings, the cross entropy calculated in (1) was used as the error function. The error function (L_total) for the entire proposed multi-task model was the sum of the error (L_ib) for the prediction of the implant brand and the error (L_ts) for the prediction of the treatment stage (2).
(1)$$L=-\sum_{i=0}t_{i}\log y_{i}$$

($t_{i}$: correct data, $y_{i}$: predicted probability of class *i*)
(2)$$L_{total}=L_{ib}+L_{ts}$$

### 2.8. Performance Metrics

Our performance metrics were precision, recall, specificity, and F1 score defined in Equations (3)–(7), respectively, which account for the relations between the positive labels of the data and those given by the classifier. We also calculated the receiver operating characteristics (ROC) curve and measured the area under the curve (AUC), which relates to a classifier’s ability to avoid false classification. In these equations, TP, FN, TN, and FP represent true positive (normal correctly identified), false positive (abnormal incorrectly identified), true negative (abnormal correctly identified), and false negative (normal incorrectly identified), respectively.
(3)$$\mathrm{accuracy}=\frac{\mathrm{TP}\;+\;\mathrm{TN}}{\mathrm{TP}\;+\;\mathrm{FP}\;+\;\mathrm{TN}\;+\;\mathrm{FN}}$$
(4)$$\mathrm{precision}=\frac{\mathrm{TP}}{\mathrm{TP}\;+\;\mathrm{FP}}$$
(5)$$\mathrm{recall}=\frac{\mathrm{TP}}{\mathrm{TP}\;+\;\mathrm{FN}}$$
(6)$$\mathrm{specificity}=\frac{\mathrm{TN}}{\mathrm{TN}\;+\;\mathrm{FP}}$$
(7)$$F1\;\mathrm{score}=2\times \frac{\mathrm{precision}\;\times \;\mathrm{recall}}{\mathrm{precision}\;+\;\mathrm{recall}}$$

### 2.9. Statistical Analysis

The differences between performance metrics were tested using chi-square analysis, using the JMP statistical software package version 14.2.0 for Mackintosh (SAS Institute Inc., Cary, NC, USA). The significance level was set to *p* < 0.05. Non-parametric tests were performed based on the results of the Shapiro–Wilk test. The difference between the multi-task and the single-task model was calculated for each performance metric using the Wilcoxon test. Effect sizes were calculated for the non-parametric tests, which are classified as follows: 0.5 is a large effect, 0.3 is a medium effect, and 0.1 is a small effect [22].

### 2.10. Visualization of Computer-Assisted Diagnostic System

CNN model visualization helps clarify the most relevant features used for classification. To identify potential correct classifications based on incorrect features, and to gain some intuition into the classification process, we identified the image pixels most relevant for classification using gradient-weighted class activation maps (Grad-CAM) [23]. Map visualizations are heatmaps of the gradients with the “hotter” colors representing the regions of more importance for classification. The heat map using Grad-CAM was reconstructed with the final convolutional layer in this study.

## 3. Results

### 3.1. Implant Brand Classification Performance

A comparison of ResNet models by model showed that the larger the number of parameters and the deeper the network, the better the accuracy. Comparing multi-task and single-task models, multi-tasking was superior in all CNNs. The highest accuracy on the ResNet 152 was 0.9908 (Table 3). The ROC curve in the multi-task models is shown in Appendix A.

### 3.2. Implant Treatment Stage Classification Performance

Similar results to the implant brand classification were obtained in the treatment stage classification. A comparison of each ResNet model by model showed that the larger the number of parameters and the deeper the network, the better the performance metrics. Comparing the multi-task and single-task models, multi-tasking was superior in all CNNs. The highest accuracy on the ResNet 152 was 0.9972 (Table 4). The ROC curve in the multi-task models is shown in Appendix A.

### 3.3. Comparison the Multi-Task and Single-Task Models in Classification Performance

We compared two groups of multi-task and single-task models for each performance metric in ResNet50. Table 5 shows the results of 30 times of 4-fold-cross validation analysis. In implant brands, the classification ability was significantly improved in all performance metrics except recall.

In treatment stages, the classification ability was significantly improved in all performance metrics. In terms of effect size, the implant brand classification was 0.4484 in accuracy, which was an effect size that could be classified as medium. The classification of the implant treatment stage was 0.8183 in accuracy, which was an effect size that could be classified to a large effect.

### 3.4. Visualization of Model Classification

Figure 3 shows an image of 12 different dental implants and treatment stages of dental implants, classified using each CNN model visualized by Grad-CAM. The single-task and multi-task models both showed an identification area that could be used to identify similar images. The implant brand focused mainly on fixtures as a feature area. On the other hand, the classification of implant treatment stages focused on healing abutments and superstructures.

## 4. Discussion

This study achieved very high performance in the classification of dental implant brands and treatment stages using CNNs. Furthermore, multi-task learning of implant brand and treatment stage classification enabled more accurate analysis with a very small number of parameters.

There have been several reports on dental implant branding studies using deep learning [24,25,26,27,28]. All of these studies were single-task, with a classification analysis performance of 0.935–0.98 for accuracy, 0.907–0.98 for recall, and 0.918–0.971 for AUC. In our study, single-task implant branding results were 0.9787–0.9851 for accuracy, 0.9726–0.9809 for recall, and 0.9996–0.9998 for AUC, similar to previous reports. Even better, our multi-task analysis showed an accuracy of 0.9803–0.9908, a recall of 0.9727–0.9886, and an AUC of 0.9997–0.9999, which were extremely high performances compared to previous reports. Accurate image classification by CNNs using dental panoramic radiographs can be more difficult than intraoral radiography [24]. Furthermore, although there were 3, 4 and 6 types of implant classifications in several previous studies [24,25,27,28], our research features 12 types of implant brand classifications, and many types could be classified. The high classification accuracy under these conditions is very meaningful.

In our previous research on dental implant classification [26], VGG was used as a CNN. In this study, we used ResNet for the implant classification and treatment stages, and the classification accuracy improved. We consider that the large amount of data used in this study and the fact that ResNet was a useful CNN for implant classification contributed to this improvement of classification performance. Interestingly, this is also the first study to classify the treatment stages of dental implants using deep learning. In this study, we classified three stages: fixture, fixture with abutment, and prosthesis, and showed much higher classification accuracy than the implant brand. Understanding the treatment stage from dental panoramic radiographs is clinically very useful for facilitating patient implant maintenance and information sharing with the dental staff.

This study was able to simultaneously classify implant brands and treatment stages. When multi-task learning correlates several tasks, the synergistic effect of learning those tasks can improve the performance of each task.

This study was able to simultaneously classify implant brands and treatment stages. When multi-task learning correlates several tasks, the synergistic effect of learning those tasks can improve the performance of each task [29]. Through multi-task transfer learning, CNNs can learn intermediate representations shared between tasks, resulting in more functional representations and better performance [17]. In fact, the difference in performance metrics between the two methods of single-task and multi-task performance was statistically significant in all but recall of the brand classification.

Another advantage is that the learning time and the total number of parameters can be reduced by solving multiple tasks with one learner [29]. This time, the total number of parameters for the two single tasks was about twice the number of parameters for the multi-task model. For example, the total number of parameters for ResNet50 was 25,659,608 for multi-tasking, 25,656,533 for brand classification, and 25,647,308 for treatment stage classification, and the total number of single-tasks was 51,303,841. Despite the small number of parameters, the classification performance was improved, and we were able to greatly benefit from the multi-task model.

In deep learning, it is necessary to prepare sufficient data for learning general-purpose parameters, but when sufficient data cannot be prepared, the learning data may be inflated to improve the recognition accuracy [30]; this is called data argumentation. On the other hand, in multi-task learning, multi-class classification is performed using a common intermediate layer, and it is possible to learn using a larger data set than when targeting a single class. If there are features common to multiple tasks in multi-task learning, the features can be regarded as performing data expansion because the learning can be performed by combining the data sets of each task [31]. In other words, the implicit data augmentation effect can be predicted. Nevertheless, it was possible to classify with high accuracy. It is considered that the common feature amount could be learned because the implant brand and the treatment stage judgment feature are in the joint part between the implant fixture and the abutment.

In this study, we were able to measure the effect size for multi-task learning. An effect size was defined as “a quantitative reflection of a magnitude of some phenomenon that is used for the purpose of addressing a question of interest” [32]. Effect size is an index that expresses the effect of experimental manipulation and the strength of association between variables. Regarding the effect size in this study, the accuracy of the implant brand classification was 0.4484, and the accuracy of the implant treatment stage classification was 0.8183; these effect sizes that could be classified as moderate and large effects, respectively. From the effect size results, the higher effect in Multitask now affects the treatment stage. It also had a moderate impact on implant brands. It is suggested that the greater contribution of Multitask was in the therapeutic phase. Our study is the first paper to show the effect size in the implant classification using deep learning, and we are confident that it will play a role as a previous study in future research. The effect sizes calculated from this experiment are useful in determining sample sizes for other studies, as there are few reports on such effect size calculations from comparisons between various deep learning models.

In this study, the accuracy of ResNet, the smallest number of parameters, was 0.980 in the brand classification and 0.9497 in the treatment stage classification. ResNet152, which has the largest number of parameters, showed extremely high accuracy, but there is a problem in practicality. In order to process all processing on the edge side by edge computing, it is necessary to consider the size of the deep learning model required for processing computational resources and tasks. In the future, it will be required to build a more accurate network with a small number of parameters. It is then desirable to export this to software available to clinicians so that dental implants can be recognized almost instantly.

This study has two limitations. First, there are several types of dental implant brands. Some implant classifications have been clarified in this study; however, they are still few, but the number of brand classifications in our study is the largest among currently published papers. Efficient classification of major dental implant brands should be the basis for classifying various types of implants around the world, including the rare implants that will be required in the future. Therefore, we believe that our research is useful; however, we expect future research to increase the number of implant brands used. Second, regarding the consideration of classification errors, this study showed very high accuracy in various CNNs. However, unfortunately, some classification errors existed even with high-precision CNNs. This study does not focus on the analysis of classification errors. In order to further improve accuracy, we would like to conduct further research with other machine learning methods, such as reinforcement learning, which will improve accuracy by using error information.

## 5. Conclusions

We have demonstrated very high performance in the classification of dental implant brands and treatment stages using CNNs in our study. Furthermore, multi-task learning of implant brand and treatment stage classification enabled more accurate analysis with a very small number of parameters. These results may play an important role in the rapid identification of dental implants in the clinical setting of dentistry.

## Figures and Tables

**Figure 1 biomolecules-11-00815-f001:**
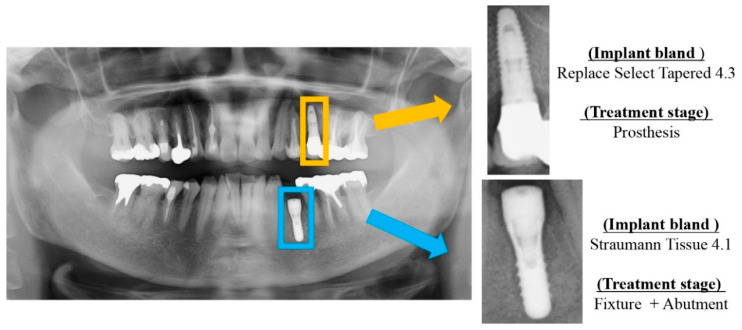
Cropping of dental implant imagery to include single fixtures. At the same time, the implant brand and treatment stage were labeled.

**Figure 2 biomolecules-11-00815-f002:**
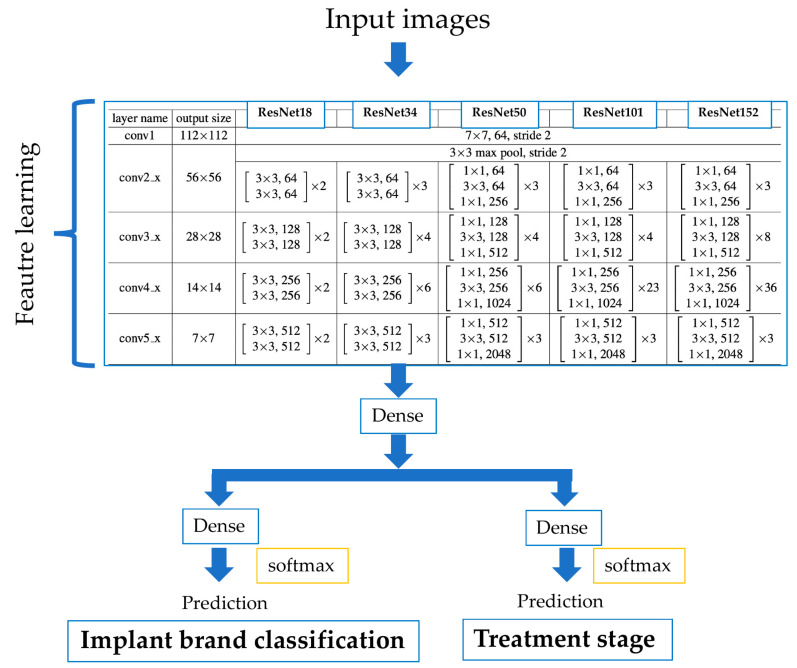
Multi-task CNN used for the implant brand and the treatment stage classifier.

**Figure 3 biomolecules-11-00815-f003:**
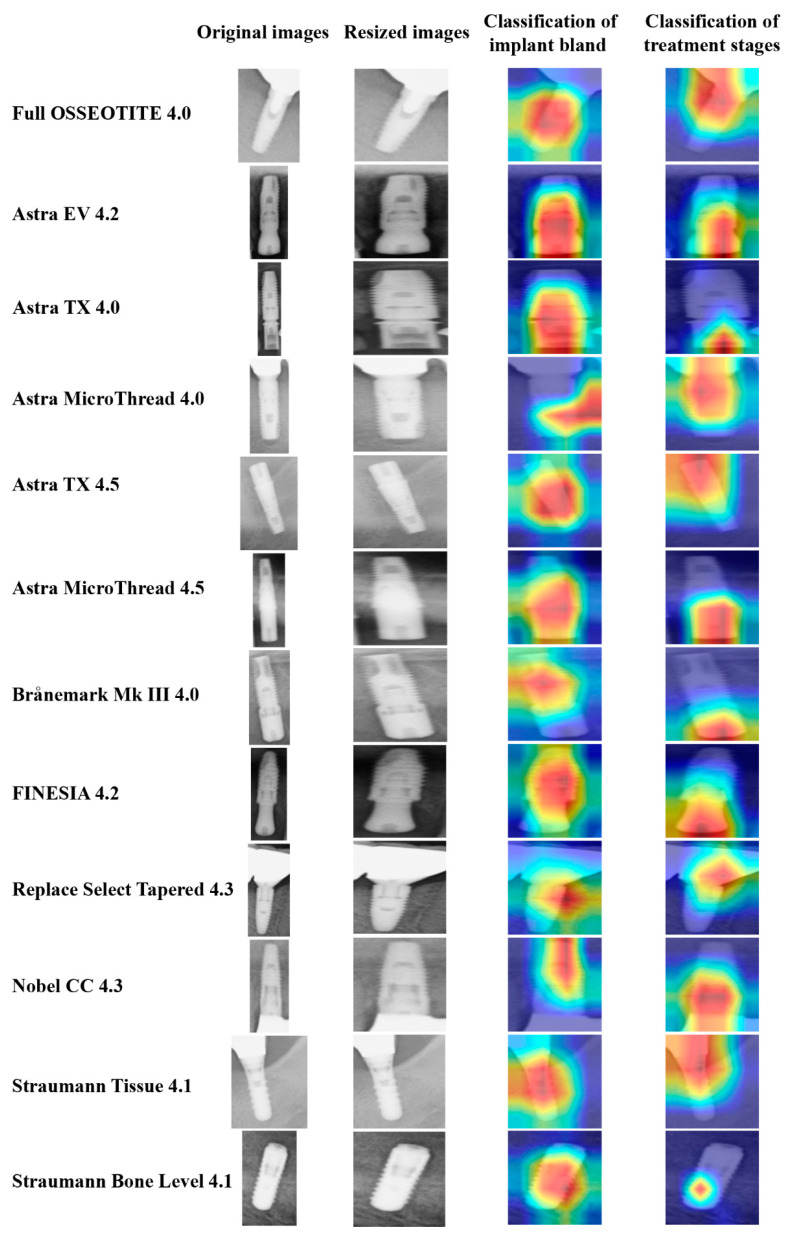
Example of the class activation maps of 12 different dental implants classified by Grad-CAM.

**Table 1 biomolecules-11-00815-t001:** Image distribution of dental implant brand system and treatment stage.

Dental Implant Bland	Treatment Status			
	Fixture	Fixture + Abutment	Prosthesis	Total
**Full OSSEOTITE 4.0**	279	25	123	427
**Astra EV 4.2**	350	307	188	845
**Astra TX 4.0**	1412	504	604	2520
**Astra TX 4.5**	523	158	433	1114
**Astra MicroThread 4.0**	337	82	285	704
**Astra MicroThread 4.5**	220	94	66	380
**Brånemark Mk III 4.0**	275	52	28	355
**FINESIA 4.2**	137	141	54	332
**Replace Select Tapered 4.3**	302	178	136	616
**Nobel Replace CC 4.3**	1089	233	277	1599
**Straumann Tissue 4.1**	225	288	142	655
**Straumann Bone Level 4.1**	87	99	34	220

**Table 2 biomolecules-11-00815-t002:** The number of parameters for each multi-task and single-task model in ResNet.

Total Parameter	ResNet18	ResNet34	ResNet50	ResNet101	ResNet152
**Multi-task**	11,457,240	21,572,824	25,659,608	44,703,960	60,393,688
**Single-task** **(Bland + Treatment stage)**	22,906,785	43,137,953	51,303,841	89,392,545	120,772,001
Bland	11,455,701	21,571,285	25,656,533	44,700,885	60,390,613
Treatment stage	11,451,084	21,566,668	25,647,308	44,691,660	60,381,388

**Table 3 biomolecules-11-00815-t003:** Dental implant brand classification performance of each ResNet CNN model.

**Multi-Task**	**ResNet18**	**ResNet34**	**ResNet50**	**ResNet101**	**ResNet152**
**Accuracy**	0.9803	0.9851	0.9869	0.9899	0.9908
**Precision**	0.9780	0.9847	0.9870	0.9895	0.9914
**Recall**	0.9727	0.9808	0.9812	0.9860	0.9886
**F1-score**	0.9749	0.9826	0.9838	0.9875	0.9899
**AUC**	0.9997	0.9998	0.9998	0.9999	0.9999
**Single-Task**	**ResNet18**	**ResNet34**	**ResNet50**	**ResNet101**	**ResNet152**
**Accuracy**	0.9787	0.9800	0.9800	0.9841	0.9851
**Precision**	0.9737	0.9790	0.9816	0.9822	0.9839
**Recall**	0.9726	0.9743	0.9746	0.9798	0.9809
**F1-score**	0.9724	0.9762	0.9776	0.9805	0.9820
**AUC**	0.9996	0.9997	0.9996	0.9997	0.9998
**Multi/Single (%)**	**ResNet18**	**ResNet34**	**ResNet50**	**ResNet101**	**ResNet152**
**Accuracy**	100.16	100.52	100.71	100.59	100.57
**Precision**	100.45	100.59	100.56	100.74	100.76
**Recall**	100.01	100.66	100.68	100.63	100.79
**F1-score**	100.25	100.65	100.64	100.71	100.80
**AUC**	100.01	100.01	100.02	100.02	100.01

(The change ratio) = (multi-task each performance metrics)/(single-task each performance metrics) ×100.

**Table 4 biomolecules-11-00815-t004:** Dental implant treatment stage classification performance of each ResNet CNN model.

**Multi-Task**	**ResNet18**	**ResNet34**	**ResNet50**	**ResNet101**	**ResNet152**
**Accuracy**	0.9947	0.9958	0.9963	0.9971	0.9972
**Precision**	0.9949	0.9958	0.9965	0.9971	0.9972
**Recall**	0.9933	0.9946	0.9949	0.9960	0.9963
**F1-score**	0.9941	0.9952	0.9957	0.9965	0.9967
**AUC**	0.9998	1.0000	0.9999	0.9999	0.9999
**Single-Task**	**ResNet18**	**ResNet34**	**ResNet50**	**ResNet101**	**ResNet152**
**Accuracy**	0.9942	0.9951	0.9960	0.9964	0.9955
**Precision**	0.9947	0.9949	0.9952	0.9970	0.9957
**Recall**	0.9922	0.9940	0.9944	0.9948	0.9943
**F1-score**	0.9934	0.9944	0.9948	0.9959	0.9950
**AUC**	0.9998	0.9999	0.9999	0.9998	0.9999
**Multi/Single (%)**	**ResNet18**	**ResNet34**	**ResNet50**	**ResNet101**	**ResNet152**
**Accuracy**	100.05	100.07	100.03	100.07	100.17
**Precision**	100.02	100.10	100.13	100.01	100.14
**Recall**	100.11	100.07	100.04	100.12	100.20
**F1-score**	100.07	100.08	100.09	100.06	100.17
**AUC**	100.01	100.01	100.00	100.01	100.00

(The change ratio) = (multi-task each performance metrics)/(single-task each performance metrics) ×100.

**Table 5 biomolecules-11-00815-t005:** Multi-task and single-task models of each performance metric in ResNet50.

**Bland Classification**						
	**Multi-task**	**SD**	**Single-task**	**SD**	***p* value**	**Effect size**
**accuracy**	0.9785	0.0123	0.9706	0.0209	0.0332	0.4484
**precision**	0.9794	0.0079	0.9724	0.0138	0.0195	0.6017
**recall**	0.9718	0.0150	0.9648	0.0200	0.0948	0.3902
**f1-score**	0.9739	0.0150	0.9662	0.0206	0.0371	0.4221
**AUC**	0.9995	0.0009	0.9989	0.0020	0.0115	0.3940
**Treatment stage**						
	**Multi-task**	**SD**	**Single-task**	**SD**	***p* value**	**Effect size**
**accuracy**	0.9963	0.0009	0.9924	0.0061	<0.0001	0.8183
**precision**	0.9961	0.0009	0.9924	0.0072	0.0004	0.6876
**recall**	0.9950	0.0013	0.9910	0.0050	<0.0001	0.9834
**f1-score**	0.9956	0.0011	0.9914	0.0063	<0.0001	0.8387
**AUC**	0.9999	0.0001	0.9997	0.0002	0.0015	0.7457

(SD; standard deviation).

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
