# Peer review of "Multi-Task Deep Learning Model for Classification of Dental Implant Brand and Treatment Stage Using Dental Panoramic Radiograph Images"

_biomolecules, 2021, doi:10.3390/biom11060815_

Round 1
Reviewer 1 Report
General comments
- This study has several strong points. It deals with a current clinical problem, as shown by reference 7. A.I. can indeed provide a solution to this problem, because it should not be too difficult for a CNN to recognize implant brands after adequate training. The study uses a large number of patients, and the results seem to indicate high performance of the CNN model. Although several previous studies on this topic has been topic, I believe the topic as such is still interesting enough.
- However, there seem to be a few general issues. First, one may wonder if implant identification will remain a problem in the future, with the electronic health records becoming ubiquitous, and many countries having a centralized healthcare system that allow for this type of info to be retrieved easily. Even in your own study, there was no need for manual labeling because all the info could be retrieved from the patient charts. Please discuss.
- Second, as the authors mentioned, there are 1000s of implant types, but the current study only includes 12 types (which is, admittedly, more than other previous studies on this topic). An A.I. system will always be difficult to train a large variety of implants, because this would require samples from each type to be collected for training. Personally, I see more potential in matching the implant outline with a database of implant CAD drawings using more conventional image processing (perhaps active shape models?). Some discussion may be needed regarding the performance of this approach in a realistic setting. Since the main issue seems to be with more ‘rare’ implant types, will an A.I. that is trained on the most common implant types really be of us?
- The article seems to have a lot of overlap with previous research by the same group. It seems that it largely uses the same sample and similar methodology, but has added one implant brand, uses a different type of CNN architecture, and added a secondary task. Although reusing data normally limits the novelty, the current study seems to show higher performance for ResNet than VGG, which is an interesting finding. However, the link and overlap with the previous research should be emphasized more, and it should be explored in more depth why this type of network seems to be more performant. The authors only make a brief mention of their own previous work in the discussion section.
- This is a typical article for a radiology or implant journal. I does not seem to have any relation with ‘biomolecules’. However, it is up to the Editor to decide if it falls within the scope of the journal
- Language is good enough for comprehension, and the manuscript is well-written and -structured. However, I believe minor review of grammar is needed.
Specific comments
- The use of two tasks is conceptually interesting, but from an outsider’s point of view, it is hard to imagine that this second task (‘treatment stage’) is truly necessary. It does not seem to serve a purpose other than to increase the performance for the primary task, and it is not of clinical interest because the treatment stage can easily be determined by the clinician. Perhaps more in-depth explanation, or additional references, is needed to explain why this second task was included. Could you have improved the performance instead by optimizing the CNN architecture and hyperparameters for the primary task? Could this primary task have been combined with a more clinically interesting secondary task (e.g. determining bone level)?
- It is always difficult to judge the performance of an A.I. system without comparing it with a human observer. The task in this case seems to be not too challenging, so perhaps a near-perfect accuracy is expected. It would make this study stronger if a calibrated human observer performs the implant identification task on the test data. Furthermore, it would be interesting to focus on the test cases with misidentification; can you determine why these were misidentified, and can you use that information to improve the CNN model further (perhaps through reinforcement learning)?
- Figure 1: ‘bland’ -> ‘brand’ (the same error is made in the text on occasion)
- “we introduced a learning method called residual learning”; please clarify if this is indeed a concept you developed, because the next sentence seems to refer to others’ work
- Is the ResNet limited to a maximum of 152 layers, and if so, why? It seems that an even deeper network could have been beneficial for this task
- I suggest showing the exact network architecture that was used rather than the simplified representation in Figure 2. The different implant brands do not need to be repeated in this figure. I would be more interested in knowing the exact number of dense nodes, and the way the CNN is structured e.g. pooling, kernel sizes, etc
- Please clarify the preprocessing. Were implants manually cropped, and then resized to 128x128 pixels? Figure 3 shows that the implants are deformed due to resizing it to an image with equal width and height. Also, the degree of deformation does not seems to be consistent because of the dimensions of the original image depending on, e.g., the implant angle. Seeing that implant length and diameter seem to be rather important features to determine the brand, doesn’t this deformation hamper the performance of the CNN?
- Section 2.6: why a fixed momentum and learning rate instead of an adaptive one, and how were the hyperparameters determined? 50 epochs seems a bit low, did you run tests with increased epochs? Training stopped if the ‘validation error did not update’: by how much?
- Combining the errors for the two tasks seems like an appropriate approach if both tasks are of equal importance, but in this case, I believe the implant brand is the main outcome. Maybe it should have been given more weight?
- L1 and L2 are functions used to express different types of loss in machine learning; using the same abbreviations for expressing the error for the two tasks in your study may lead to confusion. Perhaps different abbreviations can be used.
- Different formatting is used for equations; some are shown with letters and others with numbers. Consistent formatting is needed (and in accordance with the journal’s style)
- 5 difference outcome metrics + ROC AUC were used, which seems like overkill because of the redundancy between them (they are all derived from TP/FP/TN/FN). For a diagnostic evaluation, I would suggest Sn, Sp, and AUC. Although this is not a diagnostic evaluation, I think it is ok to use these three metrics here as well.
Author Response
Responses to Reviewers’ Comments
Thank you very much for your invaluable comments and kind acceptance. We have incorporated all the reviewers’ comments and suggestions into our manuscript; the corresponding changes are highlighted in red font in the revised manuscript.
We would like to say thank you once again for the suggestions, which were very helpful in further improving our manuscript.
Comments from Reviewers and Responses
Reviewer 1
This study has several strong points. It deals with a current clinical problem, as shown by reference 7. A.I. can indeed provide a solution to this problem, because it should not be too difficult for a CNN to recognize implant brands after adequate training. The study uses a large number of patients, and the results seem to indicate high performance of the CNN model. Although several previous studies on this topic has been topic, I believe the topic as such is still interesting enough.
Comment 1) Reviewer1: However, there seem to be a few general issues. First, one may wonder if implant identification will remain a problem in the future, with the electronic health records becoming ubiquitous, and many countries having a centralized healthcare system that allow for this type of info to be retrieved easily. Even in your own study, there was no need for manual labeling because all the info could be retrieved from the patient charts. Please discuss.
Response:
We thank you for this helpful comment. Labeling in deep learning is an important issue. It is necessary to link the image data based on the information from the patient chart. However, since implant treatment is at your own expense, it is not managed collectively by electronic medical records. Therefore, the link with the patient chart was not performed in our system, and it was necessary to manually label it.
Comment 2) Reviewer1: Second, as the authors mentioned, there are 1000s of implant types, but the current study only includes 12 types (which is, admittedly, more than other previous studies on this topic). An A.I. system will always be difficult to train a large variety of implants, because this would require samples from each type to be collected for training. Personally, I see more potential in matching the implant outline with a database of implant CAD drawings using more conventional image processing (perhaps active shape models?). Some discussion may be needed regarding the performance of this approach in a realistic setting. Since the main issue seems to be with more ‘rare’ implant types, will an A.I. that is trained on the most common implant types really be of us?
Response:
We thank you for this helpful comment. It is as you have pointed out, and that is a great idea. I think it would be very useful if we could realize a design using CAD. First, AI should be trained on the most common types of implants, and rare implants used around the world are expected to be learned using regional databases. We have added this to the Discussion section.
Comment 3) Reviewer1: The article seems to have a lot of overlap with previous research by the same group. It seems that it largely uses the same sample and similar methodology, but has added one implant brand, uses a different type of CNN architecture, and added a secondary task. Although reusing data normally limits the novelty, the current study seems to show higher performance for ResNet than VGG, which is an interesting finding. However, the link and overlap with the previous research should be emphasized more, and it should be explored in more depth why this type of network seems to be more performant. The authors only make a brief mention of their own previous work in the discussion section.
Response:
Thank you for this helpful comment. The following was added to the Discussion section. In our previous research of dental implant classification, VGG was used as CNN. In this study, we used ResNet for implant classification and treatment stages, and the classification accuracy improved. We consider that the large amount of data used in this study and the fact that ResNet was a useful CNN in implant classification contributed to the improvement of classification performance.
Comment 4) Reviewer1: This is a typical article for a radiology or implant journal. I does not seem to have any relation with ‘biomolecules’. However, it is up to the Editor to decide if it falls within the scope of the journal
Response:
Thank you for this helpful comment. We wish to submit this article for publication in the special issue "Application of Artificial Intelligence for Medical Research, 2nd Edition" of "Biomolecules."
The purpose of this issue was "With this Special Issue, we aim to cover topics on application of artificial intelligence for medical research, in particular focusing on integrated analysis of medical data using Machine Learning and Deep Learning." The study of whether osteoporosis can be classified from hip roentgen photographs is consistent with the purpose of the special issue. Thank you for your understanding.
Comment 5) Reviewer1: Language is good enough for comprehension, and the manuscript is well-written and -structured. However, I believe minor review of grammar is needed.
Response:
Thank you for this helpful suggestion. This manuscript was submitted to a native speaker for English proofreading again.
Comment 6) Reviewer1: The use of two tasks is conceptually interesting, but from an outsider’s point of view, it is hard to imagine that this second task (‘treatment stage’) is truly necessary. It does not seem to serve a purpose other than to increase the performance for the primary task, and it is not of clinical interest because the treatment stage can easily be determined by the clinician. Perhaps more in-depth explanation, or additional references, is needed to explain why this second task was included. Could you have improved the performance instead by optimizing the CNN architecture and hyperparameters for the primary task? Could this primary task have been combined with a more clinically interesting secondary task (e.g. determining bone level)?
Response:
Thank you for this helpful comment. It is as you have pointed out.
We added this task in this study because understanding the stage of treatment is clinically important. Different implant brands use different drivers for cover screws and healing abutments, and it is necessary to prepare the equipment according to the stage of treatment. Understanding the treatment stage and brand at the same time is essential for smooth treatment.
We have added the above reasons to the manuscript.
Comment 7) Reviewer1: It is always difficult to judge the performance of an A.I. system without comparing it with a human observer. The task in this case seems to be not too challenging, so perhaps a near-perfect accuracy is expected. It would make this study stronger if a calibrated human observer performs the implant identification task on the test data. Furthermore, it would be interesting to focus on the test cases with misidentification; can you determine why these were misidentified, and can you use that information to improve the CNN model further (perhaps through reinforcement learning)?
Response:
Thank you for this helpful comment.
The discrimination between people is ideal, as you pointed out, and should be a subject of further research. Furthermore, it is very interesting to note the test cases where there were errors. We actually confirmed some of the errors; however, mistakes with similar implants (different brands from the same manufacturer) and errors in judging the treatment stage due to copying of the prosthesis of the opposing tooth occurred. However, since our study was not focused on errors, we were not able to perform a statistical study at this time. We have added it to the Limitations section, warranting future study.
Comment 8) Reviewer1: Figure 1: ‘bland’ -> ‘brand’ (the same error is made in the text on occasion)
Response:
Thank you for this helpful comment.
It is our mistake. We have corrected all the words to “brand.”
Comment 9) Reviewer1: “we introduced a learning method called residual learning”; please clarify if this is indeed a concept you developed, because the next sentence seems to refer to others’ work
Response:
Thank you for this helpful comment. Residual learning is a concept that has already been reported and was used in this study. We have corrected the text.
Comment 10) Reviewer1: Is the ResNet limited to a maximum of 152 layers, and if so, why? It seems that an even deeper network could have been beneficial for this task
Response:
Thank you for this helpful comment. ResNet does not have a maximum of 152 layers. In our research, we examined up to 152 layers as a representative of the relatively commonly used ResNet. We have revised the manuscript.
Comment 11) Reviewer1: I suggest showing the exact network architecture that was used rather than the simplified representation in Figure 2. The different implant brands do not need to be repeated in this figure. I would be more interested in knowing the exact number of dense nodes, and the way the CNN is structured e.g. pooling, kernel sizes, etc
Response:
Thank you for this helpful comment. We have changed Figure 2.
Comment 8) Reviewer1: Please clarify the preprocessing. Were implants manually cropped, and then resized to 128x128 pixels? Figure 3 shows that the implants are deformed due to resizing it to an image with equal width and height. Also, the degree of deformation does not seems to be consistent because of the dimensions of the original image depending on, e.g., the implant angle. Seeing that implant length and diameter seem to be rather important features to determine the brand, doesn’t this deformation hamper the performance of the CNN?
Response:
Thank you for this helpful comment.
The implant was manually cropped and then resized to 128×128 pixels. In this study, it was transformed into a square; however, it showed very high accuracy, suggesting that length and diameter were not important clues for the classification of implant brands.
We have modified the preprocessing part in our manuscript.
Comment 9) Reviewer1: Section 2.6: why a fixed momentum and learning rate instead of an adaptive one, and how were the hyperparameters determined? 50 epochs seems a bit low, did you run tests with increased epochs? Training stopped if the ‘validation error did not update’: by how much?
Response:
Thank you for this helpful comment.
Hyperparameters were done with common values that are usually done in our lab. The results are extremely accurate, and parameter tuning was not performed in this study.
As you pointed out, 50 epochs may seem small; however, increasing the number of epochs did not change the accuracy. We also introduced early stop; yet, many trainings adopted early stop.
Comment 9) Reviewer1: Combining the errors for the two tasks seems like an appropriate approach if both tasks are of equal importance, but in this case, I believe the implant brand is the main outcome. Maybe it should have been given more weight?
Response:
Thank you for this helpful comment.
As you have pointed out, it will be necessary to investigate multitask learning with changed weights in the future. We plan to research different weight distributions further.
Comment 10) Reviewer1: L1 and L2 are functions used to express different types of loss in machine learning; using the same abbreviations for expressing the error for the two tasks in your study may lead to confusion. Perhaps different abbreviations can be used.
Response:
Thank you for this helpful comment.
Accordingly, we have changed L and L2 to L_ib and L_ts, respectively, to prevent confusion.
Comment 11) Reviewer1: Different formatting is used for equations; some are shown with letters and others with numbers. Consistent formatting is needed (and in accordance with the journal’s style)
Response:
Thank you for this helpful comment.
We have aligned the character formats.
Comment 13) Reviewer1: 5 difference outcome metrics + ROC AUC were used, which seems like overkill because of the redundancy between them (they are all derived from TP/FP/TN/FN). For a diagnostic evaluation, I would suggest Sn, Sp, and AUC. Although this is not a diagnostic evaluation, I think it is ok to use these three metrics here as well.
Response:
Thank you for this helpful comment.
We are planning to continue research for the classification of dental implants, and we will do it based on the evaluation metrics (Sn, Sp, and AUC) of the research paper presented at that time. Thank you for your valuable guidance.
Reviewer 2 Report
The study is interesting and well presented. The technical topic and form restrict the interest of the clinician. It can inspire further investigations so can be considered worthy for publication.
Author Response
Responses to Reviewers’ Comments
Thank you very much for your invaluable comments and kind acceptance. We have incorporated all the reviewers’ comments and suggestions into our manuscript; the corresponding changes are highlighted in red font in the revised manuscript.
We would like to say thank you once again for the reviewers’ suggestions, which were very helpful in further improving our manuscript.
Reviewer 2
The study is interesting and well presented. The technical topic and form restrict the interest of the clinician. It can inspire further investigations so can be considered worthy for publication.
Author response: We respect your peer-reviewed efforts for our treatises. Thank you for rating our manuscript.
Reviewer 3 Report
In this study, Sukegawa et al. proposed a deep learning model to classify dental implant brand and treatment stage using dental panoramic radiograph images. The dataset is retrospectively collected and the authors tried different transfer learning models to reach the optimal performance. There are some major points that need to be addressed in this manuscript:
1. The authors should show more literature reviews in "Introduction" about related works on deep learning-based models for dental panoramic radiograph images.
2. Preprocessing steps on images should be described clearly.
3. How did the authors tune the optimal hyperparameters of their models?
4. Why did the authors use an input shape of 128x128 in their models? Is this size optimal?
5. The authors should have validation data to evaluate the performance of the model on unseen data.
6. The authors should compare the predictive performance with previous studies on the same problem/data.
7. Source codes should be provided for replicating the methods.
8. Evaluation metrics (i.e., recall, precision, accuracy, ...) have been used in previous medical-based studies such as PMID: 32942564, PMID: 31518859, PMID: 33608558, PMID:33735760, PMID: 33806898, and PMID: 33810011, just as examples. Therefore, the authors are suggested to refer to more works in this description.
9. ROC curves should be plotted to see the performance at different threshold levels.
Author Response
Responses to Reviewers’ Comments
Thank you very much for your invaluable comments and kind acceptance. We have incorporated all the reviewers’ comments and suggestions into our manuscript; the corresponding changes are highlighted in red font in the revised manuscript.
We would like to say thank you once again for the suggestions, which were very helpful in further improving our manuscript.
Reviewer 3
In this study, Sukegawa et al. proposed a deep learning model to classify dental implant brand and treatment stage using dental panoramic radiograph images. The dataset is retrospectively collected and the authors tried different transfer learning models to reach the optimal performance. There are some major points that need to be addressed in this manuscript:
Comment 1) Reviewer3: The authors should show more literature reviews in "Introduction" about related works on deep learning-based models for dental panoramic radiograph images.
Response:
Thank you for this helpful comment.
We have cited references in the Introduction section, which point to related research on deep learning-based models of dental panoramic radiographic images.
Comment 2) Reviewer3: Preprocessing steps on images should be described clearly.
Response:
Thank you for this helpful comment. We have changed the image size as a preprocess. The contents have been added to "2.3 Data acquisition and preprocessing."
Comment 3) Reviewer3: How did the authors tune the optimal hyperparameters of their models?
Response: Thank you for this helpful comment.
Our parameters are as follows.
(Multitask)
・ Batch size: 32
・ Optimizer: momentum SGD (momentum = 0.9)
・ Learning rate: 0.001
・ Number of learning: 50 epoch
・ Loss weight (brand): 1.0
・ Loss weight (treatment stage): 1.0
(Single task (brand))
・ Batch size: 32
・ Optimizer: momentum SGD (momentum = 0.9)
・ Learning rate: 0.001
・ Number of learning: 50 epoch
(Single task (treatment stage))
・ Batch size: 32
・ Optimizer: momentum SGD (momentum = 0.9)
・ Learning rate: 0.001
・ Number of learning: 50 epoch
We did not tune it because it was accurate enough.
Comment 4) Reviewer3: Why did the authors use an input shape of 128x128 in their models? Is this size optimal?
Response: Thank you for this helpful comment.
Larger image sizes can generally increase accuracy. This study used 128×128 pixels to show sufficient accuracy. Therefore, we consider it to be an appropriate size.
Comment 5) Reviewer3: The authors should have validation data to evaluate the performance of the model on unseen data.
Response: Thank you for this helpful comment.
In this research, deep learning was performed by dividing the data for verification into training data and verification data in an 8 : 2 ratio from the data excluding the test data. The contents of "2.5. Model training" have been revised.
Comment 6) Reviewer3: The authors should compare the predictive performance with previous studies on the same problem/data.
Response: Thank you for this helpful comment.
We have added a discussion of comparisons with our previous study. We have added the different numbers of data and the differences between the CNN models.
Comment 7) Reviewer3: Source codes should be provided for replicating the methods.
Response: Thank you for this helpful comment.
We will refrain from publishing the source code for this study for now because it is necessary to discuss with the Intellectual Property Department.
Comment 1) Reviewer3: Evaluation metrics (i.e., recall, precision, accuracy, ...) have been used in previous medical-based studies such as: PMID: 32942564, PMID: 31518859, PMID: 33608558, PMID:33735760, PMID: 33806898, and PMID: 33810011, just as examples. Therefore, the authors are suggested to refer to more works in this description.
Response: Thank you for this helpful comment.
We are planning to continue research for the classification of dental implants, and we will do it based on the evaluation metrics (Sn, Sp, and AUC) of the research paper presented at that time. Thank you for your valuable guidance.
Comment 1) Reviewer3: ROC curves should be plotted to see the performance at different threshold levels.
Response: Thank you for this helpful comment.
We have added the ROC curve in Supplementary Materials.
Round 2
Reviewer 3 Report
My previous comments have been addressed well.